# Functionalized Nanomaterials in Cancer Treatment: A Review

**DOI:** 10.3390/ijms26062633

**Published:** 2025-03-14

**Authors:** Oscar Gutiérrez Coronado, Cuauhtémoc Sandoval Salazar, José Luis Muñoz Carrillo, Oscar Alexander Gutiérrez Villalobos, María de la Luz Miranda Beltrán, Alejandro David Soriano Hernández, Vicente Beltrán Campos, Paola Trinidad Villalobos Gutiérrez

**Affiliations:** 1Centro Universitario de los Lagos, Universidad de Guadalajara, Lagos de Moreno 47460, Mexico; ogutierrez@culagos.udg.mx (O.G.C.); delaluz.miranda@academicos.udg.mx (M.d.l.L.M.B.); gigiouxy@hotmail.com (A.D.S.H.); 2División de Ciencias de la Salud e Ingenierías, Campus Celaya-Salvatierra, Universidad de Guanajuato, Celaya 38060, Mexico; cuauhtemoc.sandoval@ugto.mx (C.S.S.); vbeltran@ugto.mx (V.B.C.); 3Facultad de Medicina, Centro Universitario de Ciencias de la Salud, Universidad de Guadalajara, Guadalajara 44340, Mexico; ozkarlex@gmail.com

**Keywords:** cancer, functionalized nanoparticles, metallic nanoparticles, organic nanoparticles, carbon-based nanomaterials

## Abstract

Cancer is one of the main causes of death worldwide. Chemotherapy, radiotherapy and surgery are currently the treatments of choice for cancer. However, conventional therapies have their limitations, such as non-specificity, tumor recurrence and toxicity to the target cells. Recently, nanomaterials have been considered as therapeutic agents against cancer. This is mainly due to their unique optical properties, biocompatibility, large surface area and nanoscale size. These properties are crucial as they can affect biocompatibility and uptake by the cell, reducing efficacy. However, because nanoparticles can be functionalized with biomolecules, they become more biocompatible, which improves uptake, and they can be specifically targeted against cancer cells, which improves their anticancer activity. In this review, we summarize some of the recent studies in which nanomaterials have been functionalized with the aim of increasing therapeutic efficacy in cancer treatment.

## 1. Introduction

Cancer is one of the main causes of death in the world and the second leading cause worldwide. In 2022, 20 million new cases were diagnosed, and 9.7 million patients died that year. It is estimated that the number of cancer cases will rise to 35 million worldwide by 2050 [1]. Most cancer-related deaths are expected to occur in metastatic disease [2]. Cancer is a multifactorial disease in which genetic factors, environmental factors and an inappropriate diet play a role. The presence of these risk factors leads to mutations within the cell, resulting in uncontrolled growth and the spread of abnormal cells to other organs in the body [3]. Cancer is a group of diseases that are acquired through changes at the DNA level that lead to uncontrolled cell growth. It is believed that the vast majority of cancer cell genotypes are due to six alterations in cell physiology that together determine malignant growth. These changes are self-sustained growth signaling, insensitivity to growth-inhibitory signals, evasion of programmed cell death, limitless replication potential, persistent angiogenesis and tissue invasion and metastasis. Each of these changes in cell physiology represents a change in the defense mechanism that every cell has to fight cancer [4]. The rapid proliferation of tumor cells is the main feature of carcinogenesis. Together with the evasion of cell checkpoints that are essential for cell division, these are the mechanisms that promote the aggressive growth of tumor cells [5].

Conventional cancer treatment currently includes surgery, radiotherapy, chemotherapy and combinations of these procedures [6,7]. In addition, immunotherapy has developed rapidly and made significant advances with the aim of improving survival and prognosis in some cancers and preventing recurrence [8]. However, conventional treatments also have their limitations, such as recurrence in solid tumors, high systemic toxicity in healthy organs, drug resistance, cell escape from apoptosis, cell cycle changes and altered DNA damage response or enhanced damage repair [9].

Nanotechnology has made considerable progress in the development and delivery of drugs, so that the use of nanomaterials in cancer treatment can be more specific [10]. The delivery of chemotherapeutic agents can be improved by nanomedicine as absorption, bioavailability and release patterns can be controlled when the drugs are loaded in nanomaterials [11]. In addition, surface decoration of nanoparticles (NPs) with some molecules, such as peptides, eptamers, polymers, antibodies and other biomolecules, can target biomarkers overexpressed by tumor cells to improve specificity in the tumor. This specificity means a reduction in the secondary or toxic effects that occur in healthy organs or tissues [12]. Another special feature of nanomaterials is that nanoformulations can be synthesized by physical encapsulation or chemical conjugation with hydrophobic and hydrophilic anticancer drugs, resulting in a synergistic anticancer effect between them [13]. These nanomaterials include polymer conjugates, polymeric NPs, lipid-based carriers such as liposomes and micelles, dendrimers, carbon nanotubes and metallic NPs [14]. By endowing these nanocarriers with smart properties, we can use them to develop smart anticancer drug delivery materials that respond to endogenous and/or exogenous stimuli such as pH, reactive oxygen species (ROS), glutathione (GSH), hypoxia, enzymes, temperature, light, ultrasound, radiation and magnetic fields [15]. In this review, we show the research work that has been carried out in the field of nanomedicine. In particular, we focus on the impact that the functionalization of metallic, organic NPs and carbon-based nanomaterials has on cancer therapy.

## 2. Nanomaterials

Nanomaterials can be defined as materials that are measured on the nanometer scale from 1 to 100 nm. They can be unbound particles or particles in a state of aggregation with one or more dimensions [16]. Their unique properties are based on their size on the nanometer scale and their high surface/volume ratio [17]. These structures can be designed with a wide variety of chemical and structural compositions and thus have very different surface, magnetic, electronic and optical properties [18].

Based on their shape, nanomaterials can be classified as nanoparticles, nanocrystals, nanowires, nanorods, nanospheres, nanoprisms, nanorings and thin films [19]. However, they can also be categorized into four types based on their structural dimensions, including zero-dimensional (0D), one-dimensional (1D), two-dimensional (2D) and three-dimensional (3D) configurations [20] (Figure 1). In the 0D configuration, the nanoscale can be observed in the x, y and z axes along all three dimensions and consists of nanoparticles, quantum dots and nanospheres. In 1D nanomaterials, they expand in one direction, such as nanowires, nanorods and nanotubes. In 2D nanomaterials, they grow in both the x and y directions, while the third dimension remains in the nanometer range, e.g., nanofilms and nanosheets. Finally, 3D nanostructures grow in their three dimensions, and all dimensions of these nanostructures are outside the nanometer range or larger than 100 nm, as the nanomaterials of lower dimensions are ordered as blocks to give 3D structures, e.g., multi-nanolayers, bundles of nanorods or nanowires, or ordered aggregates of NPs [21,22].

A further classification is based on their structural composition. In this sense, NPs can be divided into four groups: organic nanomaterials, inorganic nanomaterials, carbon-based nanomaterials and composite nanomaterials [23]. Organic NPs are mainly synthesized from organic molecules. These NPs include liposomes, dedrimers, micelles and ferritin. Micelles and liposomes are biodegradable NPs, so they are not considered toxic. They have a hollow interior and are sensitive to electromagnetic radiation and heat [24]. In addition, these NPs are formed by non-covalent intermolecular interactions, which makes them more sensitive to degradation and facilitates their elimination in a living system. Some parameters that determine the range in which they can be used are, for example, composition, surface morphology, stability and load-bearing capacity [24]. Inorganic NPs are nanomaterials that lack carbon atoms but are composed of metals or metal oxides and are considered more stable than organic NPs [25]. Metallic NPs can be synthesized from metals, and the mechanism by which these NPs are obtained can be constructive or destructive. These NPs are produced exclusively from metallic starting materials. The metals commonly used for the synthesis of these NPs can be aluminum (Al), cadmium (Cd), cobalt (Co), copper (Cu), gold (Au), iron (Fe), lead (Pb), silver (Ag) and zinc (Zn) [26]. Metallic NPs have a high surface-to-volume ratio and quantum effects, which gives them special properties, such as sensitivity in the ultraviolet and visible range, as well as electrical, catalytic and thermal properties. These NPs have unique optoelectrical properties due to localized surface plasmon resonance (LSPR) and are used in various fields of medicine due to their special properties [27,28]. Metal oxide NPs consist of positive metal ions and negatively charged oxygen ions. This ionic interaction is strong and stable due to the electrostatic interaction resulting from the positive metal ions and the negative oxygen ions. These NPs are used in various fields, e.g., as fluorescence and optical sensors, catalysts, in photovoltaics and in biomedicine. These applications are related to their size and shape, which mainly influence their optical, mechanical and electrical, magnetic and catalytic properties [29,30].

Carbon-based nanomaterials are made of carbon, one of the elements that has the ability to form bonds with other carbons as well as with other materials such as metals, metal oxides and polymers. These nanomaterials combine the properties that characterize sp2-hybridized carbon bonds with the physicochemical properties that result from the nanometer scale. They therefore exhibit electrical conductivity, high strength, electron affinity, structure, thermal and sorption properties [28,31]. This enables them to be used in a wide range of applications, including drug delivery, energy storage and biological applications. These nanomaterials are structured in different ways, e.g., fullerenes, carbon nanotubes, carbon nanofibers, carbon quantum dots, graphene and many others [32,33].

## 3. Therapeutic Role of Nanomaterial in Cancer

Cancer treatment with nanomaterials has a major impact due to the advantages it offers over conventional therapy with free drugs, as the use of nanomaterials in drug delivery can be more targeted, which means less toxicity as well as a reduction in side effects and an increase in half-life [34].

### 3.1. Metallic Nanoparticles

Metallic NPs can have different shapes depending on the synthesis conditions, e.g., spherical, rod-shaped, streamlined, film-shaped and others. When the morphology, size and composition change, their physical and chemical properties also change [35]. Noble metals such as Au, Pt and Ag are considered biocompatible materials, mainly due to their high corrosion resistance, chemical inertness and low cytotoxicity. Due to their biocompatibility and surface functionality, these NPs can be used in various biomedical applications [36,37]. For example, Au, Pt and Ag NPs can be used as therapeutic agents against cancer due to their improved tunable optical properties (Table 1). In addition, the therapeutic efficacy of drugs can be increased as they can be loaded onto the NPs. And the surface can be modified by functionalization with biomolecules linked by hydrogen bonds, covalent bonds and electrostatic interactions [38].

All inorganic NPs have a specific core/shell structure, where the core contains metals that determine the fluorescence, optical, magnetic and electronic properties of the particle. The shell usually consists of metal or organic polymers that protect the core from interactions with the external environment and serve as a substrate for the functionalization of the NPs with biomolecules such as proteins, polyunsaturated and saturated fatty acids, peptides, nucleic acids, polysaccharides, antibodies, tumor markers and small molecules [39]. By adding molecules to the surface of the NPs, the treatment becomes more specific without damaging healthy organs. The size and shape of the NPs can also be controlled to optimize therapeutic efficacy. These NPs have a high surface-to-volume ratio and can therefore be easily modified for targeted cancer therapy [40].

#### 3.1.1. Gold

Gold is one of the precious metals characterized by its resistance to corrosion and oxidation. These properties turn this element into gold nanoparticles (AuNPs), which are used in many different applications. AuNPs exhibit unique physicochemical properties, including surface plasmon resonance (SPR) and their ability to bind amine and thiol groups, which enables their surface modification and application in the biomedical field [41,42]. The functionalization of these NPs is currently the subject of much research, as it can have an important effect in two respects: firstly, it prevents the aggregation of AuNPs, and secondly, it improves bioavailability. As a result, the AuNPs have specific interactions with the target cells, improve transport and specific accumulation in the desired organ, which can improve cancer therapy [43,44]. Several studies have shown the effect that functionalized AuNPs have in the photothermal therapy of cancer. For example, a recent study showed the effect of AuNPs functionalized with 2-thiouracil (AuNPS-2-TU) and irradiated with 520 nm light on a hormone-independent breast cancer cell line (MDA-MB-231). The effect of AuNPs-2-TU upon irradiation was an antiproliferative effect that the functionalized NPs exerted on cancer cells compared to non-irradiated NPs. This demonstrated a synergistic effect between photothermal therapy and 2-thiouracil, enhancing the anticancer effect of the drug [45]. In another study, AuNPs functionalized with HER-2 were developed for targeted therapy with dasatinib (AUNPs-HER-2-DSB) in combination with radiotherapy. This material was tested on breast cancer cells (BT-474 and MCF-7), where the interaction of AuNPs-HER-2-DSB with cancer cells showed a cytotoxic effect, and this effect was favored when the cells and the material were irradiated, showing a better cytotoxic effect as well as increased cell uptake. This was mainly due to the fact that HER-2 interacted with its receptor and thus limited the effect of dasatinib on the cancer cells [46]. A similar effect was shown by conjugating trastuzumab (Tmab) to AuNPs (T-AuNPs) on HER2-positive Tmab-resistant (MKN7) and Tmab-sensitive (NCI-N87) gastric cancer cell lines and subcutaneous tumors in vivo. The cytotoxic effect of T-AuNPs was mainly mediated by autophagy, as MNK7 and NCI-N87 cells treated with the NPs exhibited strong LC4-II expression. Furthermore, it was observed that a stronger oxidative stress was induced only in MKN7 cells. These results indicate that autophagy is the mechanism by which the T-AuNPs exert their cytotoxic effect, while oxidative stress varies depending on the cell type. On the other hand, intratumoral injection of T-AuNPs suppresses tumor growth with an increase in the autophagy marker LC3-II, suggesting that the antitumor effect is also due to the induction of autophagy [47]. Another mechanism by which cell death can be induced is apoptosis. On this basis, AuNPs conjugated with anti-HER2 antibodies (HER2-AuNPs) were examined on G361 melanoma cells. A cytotoxic effect of the functionalized NPs was found, which was mainly triggered by apoptosis. An increase in the expression of caspase-3 as well as caspase-9 level was detected; in addition to this effect, HER2-AuNPs were observed to downregulate cyclin-dependent kinase 4 (cdk4), which is related to the G1 phase, while in the G2/M phase, cyclin A, cyclin E, CDK2 and cdc2 were also decreased, suggesting that HER2-AuNPs are also related to cell cycle arrest [48].

#### 3.1.2. Platinum

Platinum is a noble metal and the first metal to prove effective in cancer therapy, especially in testicular, ovarian, bladder, cervix, skull base, larynx and lung cancer (small cell and non-small cell). However, these platinum-based drugs have shown toxic effects at the kidney, brain, nerve tissue and bone marrow levels, leading to side effects [49]. Some studies have shown that platinum nanoparticles (PtNPs) themselves have an anticancer activity. These NPs behave differently from platinum-containing compounds but have an equally efficient anticancer activity. They can increase the effectiveness of antitumor therapy. PtNPs enter the cell by passive diffusion or endocytosis. Once inside the cell, they exert their cytotoxic effect, which depends on their size, concentration and incubation time. The effect causes DNA damage leading to replication inhibition, cell cycle arrest and apoptosis [50]. However, the systemic toxicity of platinum-based drugs may limit their use. Therefore, various surface modifications of PtNPs have been proposed to improve the stability and biocompatibility of NPs. Among these modifications, biocompatible polymers such as polyethylene glycol (PEG), chitosan, xanthan gum and others stand out. Functionalization of PtNPs with antibodies, peptides and DNA/RNA has also been proposed to improve therapeutic efficacy against cancer [51].

Nanoparticle-based drug delivery systems (DDS) are considered important for the development of targeted drug delivery as well as prolonged circulation, superior bioavailability, controlled and sustained release of drugs. These systems can prevent damage to the surrounding organs, which significantly reduces the side effects due to the effect of the administered drugs [52]. On this basis, an in vitro (A549: human lung cancer cell line) and in vivo (B16F10: mouse melanoma cell line) study investigated PEG-coated PtNPs used as vehicles for doxorubicin (PtNPs-DOX). The anticancer activity was observed by a decrease in cell viability in both cancer cells mediated by apoptosis, as well as a slowing of the cell cycle at different stages (sub-G1 and G2/M), ultimately leading to cancer cell death. In the in vivo model, treatment with PtNPs-DOX showed upregulation of p53 and caspase-3 protein, while Bcl-2 expression was downregulated, suggesting that the tumor cells enter an apoptosis process, which was the main mechanism exerted by PtNPs-DOX in the in vivo melanoma model [53]. Another study showed the effect of doxorubicin-conjugated PtNPs in octopod form (OcPtNPs-DOX) on MCF-7 and MDA-MB-231 cells. The conjugated NPs had increased cytotoxicity, which was mainly due to the sustained release of doxorubicin from the PtNPs. The cytotoxicity mechanism exerted by the nanosystem was apoptosis. This effect was mediated by the mitochondrial dysfunction and activation of caspases-3 and -9 [54]. The combination of treatments such as chemotherapy and photothermal therapy could increase the therapeutic efficacy of drug-resistant breast cancer cells. On this basis, a nanoplatform based on mesoporous PtNPs with a PEG-modified surface was developed and loaded with doxorubicin (PEG@Pt/DOX). This nanoplatform was tested on MCF-7/ADR cells and showed a cytotoxic effect when only the PEG@Pt/DOX was tested. However, this effect was enhanced when this nanosystem was combined with photothermal therapy, suggesting that there is a synergistic effect between chemotherapy and photothermal therapy in the treatment of cancer [55].

PtNPs are known to cause minimal damage to surrounding cells or tissues. NPs that enter the bloodstream can be recognized by proteins that form NPs-protein complexes, which are recognized by the phagocytic system and eventually removed from the bloodstream [56]. However, in order to improve their biocompatibility and circulation lifetime, these NPs can be functionalized on their surface by biopolymers. In this sense, the synthesis of chitosan-stabilized PtNPs (Ch-PTNPs) was carried out in a study and evaluated in breast cancer cell lines (MDA-MB-231). The Ch-PTNPs reduced the viability of breast cancer cells. This cytotoxic effect was mainly related to the fact that the Ch-PTNPs induced apoptotic cell death, which was mainly due to the activation of apoptotic endonuclease. Chromatin condensation and brilliant nuclear fragmentation were also observed. The targeted effect that Ch-PTNPs exerted on cancer cells could mainly be attributed to the fact that these cells rapidly internalized the coated nanomaterials and thus exerted their anticancer effect [57]. In addition, another study investigated these NPs (Ch-PTNPs) in MCF7 cells and showed an antiproliferative effect, suggesting that this type of coated nanomaterials has an anticancer effect [58]. Surface modification with PEG is used to improve the properties of drug delivery systems. In addition, some ligands such as antibodies could be used to enable more specific and targeted therapy. In one study, PtNPs were encapsulated with poly (lactic-co-glycolic acid) (PLGA), and PEG was added, and finally, targeting was activated with an anti-EGFR ligand (PTNPs-PLGA-PEG-EGFR). These were investigated in triple-negative breast cancer cells (MDA-MB-231). Surface modification with active targeting showed an increase in the internalization of PTNPs-PLGA-PEG-EGFR, so that the cytotoxic effect of these NPs was more effective than that of NPs without anti-EGFR, suggesting that PTNPs-PLGA-PEG-EGFR are a more effective therapy and an alternative in the treatment of cancer [59]. It is known that some tumor cells overexpress receptors for hyaluronic acid (HA). One study showed the effect of PtNPs encapsulated in hyaluronic acid on MDA-MB-231 cells. These cells expressed the receptor for HA, so the accumulation of these NPs was greater compared to cells that did not express receptors for hyaluronic acid; moreover, when photothermal therapy was applied, a cytotoxic effect was exerted that targeted cells expressing the receptor for hyaluronic acid. In addition, the administration of PtNPs encapsulated in hyaluronic acid in the in vivo model was able to detect the tumor, and photothermal therapy reduced the size of the tumor, which was due to an increase in the temperature of the tumor, suggesting specificity of the treatment [60].

#### 3.1.3. Silver

Among metallic nanomaterials, silver nanoparticles (AgNPs) stand out due to their wide range of applications in various fields, including nanomedicine [61]. AgNPs can be synthesized according to the biological activity they are intended to exert, so that these NPs can be designed and produced with specific properties by controlled synthesis methods [62]. The biological activity of AgNPs depends on several factors, including surface, size, shape, morphology, composition, reactivity, coating/capping efficiency of ion release and cell type. These factors are crucial for the cytotoxic capacity of AgNPs as they can influence cellular uptake and biodistribution [63]. The properties of AgNPs make it easy to functionalize them. Surface modification is of great importance as this modification can reduce toxicity, prevent aggregation and improve the ability to target specific cells, especially cancer cells [64]. In a study, AuNPs functionalized with glucose (G-AgNPs) were developed and their cytotoxicity in prostate cancer cells (hormone-resistant cells DU-145 and PC-3) and hormone-sensitive cell lines (LNCaP) was investigated. When cancer cells were treated with G-AgNPs, only DU-145 and PC-3 cells showed a cytotoxic effect. This was mainly due to the Warburg effect, as these cells had low O_2_ consumption, higher lactate production and higher glucose consumption, which favored the uptake of G-AgNPs in DU-145 and PC-3 cells. Once the G-AgNPs were inside the cells, lysosomal enzymes released Ag^+^ and induced the release of ROS, resulting in oxidative damage to mitochondria, which, in turn, led to DNA fragmentation and eventually apoptosis, and cell cycle arrest in S phase was also observed. This suggests that these NPs exert their cytotoxic effect on hormone therapy-resistant prostate cancer cells through these mechanisms [65].

Capping of AgNPs with different polymeric materials can modulate their physicochemical properties, biodistribution, nanoparticle–cell interaction, cellular uptake and intracellular Ag^+^ release, which may affect the cytotoxic effect on cancer cells [66]. In one study, a combination of an anticancer drug (doxorubicin) was formulated on AgNPs and three different polymeric shells: polyvinyl alcohol (PVA), PEG and polyvinylpyrrolidone (PVP). The cytotoxic effect of the different systems was investigated in vitro on breast cancer (MCF-7) cell lines. AgNPs coated with PVA, PEG and PVP and loaded with doxorubicin had a greater selectivity for cancer cells, inducing a cytotoxic effect. In particular, those coated with PVP induced greater cytotoxicity on MCF-7 cells. This suggests that the surface coating of AgNPs directly influences their cytotoxic effect. This effect was likely mediated by the generation of ROS that cause oxidative stress and lipid peroxidation, leading to DNA damage and eventually cancer cell death [67]. In addition, the impact of coating AgNPs with the carbohydrate-based biopolymer chitosan (CHI-AgNPs) on breast cancer cells (MCF-7) was investigated. Treatment with these NPs showed a strong and selective intrinsic cytotoxic effect on cancer cells. This effect may be related to the fact that chitosan can modulate intracellular Ag^+^ release, which accumulates in the nucleus and leads to genotoxicity and mitochondrial damage, which together can induce apoptosis in breast cancer cells [68]. A similar study showed the same effect, but in this study, the antiproliferative effect of CHI-AgNPs on MCF-7 cells was concentration-dependent [69].

AgNPs can be used in photodynamic applications by generating ROS when irradiated with a near-infrared (NIR) laser [70]. In this context, a prodrug system with high selectivity was developed. This system consisted of AgNPs-PEG decorated with a folic acid (FA)-targeting ligand and conjugated with doxorubicin. It was tested on cells overexpressing the folic acid receptor (SKOV3 and L1210) and on cells not expressing this folate receptor (MES-SA and MES-SA/Dx5). The prodrug system was found in a larger proportion in cells expressing the folate receptor. Entry into these cells occurs via the folate receptors, so that once inside, the doxorubicin is released and exerts its cytotoxic effect on SKOV3 and L1210 cells. The photodynamic effect of the prodrug system on the cells was also investigated. Exposure to a NIR laser increases the toxicity of the NPs. This effect is mainly due to the generation of ROS. This suggests that photodynamic therapy increases the therapeutic efficacy of the prodrug system [71]. In addition, one study developed AgNPs conjugated with an epidemic growth factor receptor-specific antibody (anti-EGFR) that targets cancer cells with overexpression of EGFR, so that these anti-EGF-conjugated AgNPs (anti-EGFR/AgNPs) can be used for targeted anticancer treatment. The NPs were examined in epithelial cells of nasopharyngeal carcinoma (NEC). The results showed an inhibition of proliferation of the human NEC cell line. In addition, anti-EGFR/AgNPs was an effective radiosensitizer that also showed downregulation of various DNA repair proteins, which eventually induced apoptosis in the cells after irradiation [72]. In addition, cancer therapy must be specific, which leads to the accumulation of NPs in cancer tissue. The specificity can be mediated by specific ligands that the cancer cell can recognize; in this sense, tumor cells showed an overexpression of receptors for IgG, so with the design of AgNPs functionalized with IgG molecules (IgG-AgNPS), these NPs were administered to pancreatic cancer cell lines (Panc-1), and laser-induced hyperthermia therapy was applied. The cytotoxic effect of IgG-AgNPS and photothermal therapy is based on a disruption of the Golgi apparatus, which leads to an activation of the apoptotic caspase-3 pathway, resulting in a more selective apoptosis of the cancer cells thanks to the antibody-conjugated NPs [73]. This shows that therapy with AgNPs functionalized with polymers, specific antibodies and/or drugs has promising effects in cancer therapy, even against drug-resistant cancers.

### 3.2. Organic Nanoparticles

Polymeric nanomaterials can be widely used in biomedicine, e.g., in tissue regeneration and the controlled release of drugs and tumor immunotherapy, due to their low toxicity, high biodegradability, large surface area and ease of structural modification. These nanomaterials include polylactic co-glycolic acid (PLGA), PEG, poly (hydroxyacetic acid), polycaprolactone, polyurethane, lipid–polymer nanoparticles, lipid polymeric hybrid nanoparticles, metal–organic frameworks, and others, as shown in Table 2 [74].

#### 3.2.1. PLGA-Based Nanoparticles

PLGA particles are the most widely used type of nanoparticle, mainly because they exhibit favorable degradation as they undergo hydrolysis in the body, producing lactic and glycolic acid monomers; they also have a high biocompatibility [75,76]. The physicochemical properties, biodegradation rate and in vivo behavior of PLGA nanoparticles can be modified depending on the preparation method and molecular weight used [77]. These NPs have been evaluated as drug delivery systems, e.g., for antibiotics, antiseptic, anti-inflammatory, antioxidant and chemotherapeutic drugs. These systems may have advantages for a more specific treatment against cancer [78].

In cancer therapy, PLGA nanoparticles are used to encapsulate anticancer drugs because they have controlled and sustained release, high loading capacity and superior stability to enable specific release in the target tissue [76,79]. However, several anticancer drugs do not differentiate between normal and cancer cells. Therefore, surface modification of NPs is a strategy to make them more specific and thus achieve their target [80]. Functionalizing the surface of a nanosystem with a cell-specific ligand that has the ability to bind and specifically recognize a target cell releases the drug only in the cell population where the ligand–receptor interaction occurs. Recently, the functionalization of NPs with antibodies has led to promising therapeutic results in the therapy of some types of cancer. For example, in the treatment of gastric cancer, PEG-PLGA nanoparticles were modified with anti-CD133 and the drugs methioninase (METase) and pemetrexed were combined and encapsulated in these NPs to test them on gastric cancer cells (CD133+ SGC7901 and MKN45). Treatment of cancer cells with drug-loaded and functionalized NPs showed an increase in apoptosis in CD133+ SGC7901 cells as well as cleaved caspase 3 protein levels and an inhibition of DNA synthesis. These results suggest that this nanostructured system could increase the therapeutic efficacy of drugs in cancers overexpressing CD133 [81]. In another study, oxaliplatin (OXA) encapsulated in PLGA-based nanoparticles functionalized with anti-CD133 (PLGA-OXA-Ab) was investigated for the treatment of colorectal cancer (Caco-2). This showed a cytotoxic effect as well as a high efficiency in binding PLGA-OXA-Ab to Caco-2 cells overexpressing CD133 to achieve a more selective and specific drug release [82].

Anticancer drugs are not always water soluble, resulting in low bioavailability and therapeutic efficacy. In this sense, organic NPs offer a solution to this problem. In one study, lipid–PLGA encapsulated with salinomycin functionalized with a CD44 antibody (SM-LPN-CD44) was developed and investigated in CD44+ prostate cancer cells. SM-LPN-CD44 showed a cytotoxic effect on cells overexpressing CD44, suggesting that targeted drug delivery via anti-CD44-functionalized NPs is more selective for cancer cells possessing the ligand for this antibody [83]. An important effect of NPs is that they can be customized for each type of cancer. In particular, triple-negative breast cancer (TNBC) is an aggressive type of cancer as it does not express estrogen receptors (ER), progesterone receptors (PR) or the human epidermal growth factor receptor (HER2). Consequently, there is no standardized treatment regimen for TNBC due to the absence of these receptors [84]. In the treatment of this type of aggressive cancer, PLGA-NPs loaded with paclitaxel (PTX) (NP-PTX) and anti-EGFR protein-anchored NPs+PTX (EGFR-NP-PTX) were developed, and these NPs were tested in vitro on the MDA-MB-468 TNBC cell line and in vivo. The NPs showed a cytotoxic effect in vitro, which was mainly due to apoptosis, as condensed chromatin, fragmented nucleus and the formation of apoptotic bodies could be observed. The tumor volume was also significantly reduced in the in vivo model, especially with EGFR-NP-PTX. In addition, a high distribution in the tumor was observed with these NPs, indicating a superior cytotoxic efficiency in the tumor [85]. In another study, the effectiveness of drug encapsulation and its functionalization with antibodies was demonstrated. For this study, doxorubicin (DOX) was encapsulated in PLGA and functionalized with antibodies against Frizzled7 (DOX-PLGA-FZD7). These NPs were examined in the MDA-MB-468 TNBC cell line. These cells were exposed to DOX-PLGA-FZD7, releasing DOX into the cytoplasm of the cell and exerting its effect at the DNA level. This effect caused cytotoxicity, which drove the cell to apoptosis and/or necrosis. This specific cytotoxic effect was mainly due to the fact that the cell line MDA-MB-468 overexpressed the Frizzled7 receptor, and DOX-PLGA-FZD7 therefore bound specifically to the cancer cells (Figure 2). This shows that the functionalization of NPs produces a therapy with greater specificity and thus has therapeutic efficacy of the drugs that can be encapsulated in the different types of organic NPs [86].

#### 3.2.2. Lipid Nanoparticles

Other nanomaterials are lipid–polymer nanoparticles (LPNPs), which are excellent for drug delivery, either by immediate or sustained release. Due to their nature, these NPs are less toxic, and they have greater biocompatibility and higher stability than inorganic or polymeric nanoparticles; they also have a high capacity to incorporate hydrophilic and lipophilic compounds [87,88]. For these reasons, LPNPs are considered as an alternative for nanodelivery systems in cancer therapy. These NPs can encapsulate the currently used chemotherapeutic agents and thus avoid nonspecific distribution leading to undesirable effects that occur with these agents [89]. These NPs have already been studied in some types of cancer. Extending this approach, a chemotherapeutic formulation of methotrexate-loaded surface-modified LPNPs (Met-LPNPs) was investigated in human breast cancer (MCF-7) and human lung cancer (A549). Treatment with Met-LPNPs showed a more effective cytotoxic effect through the induction of apoptosis [90]. In addition, the chemotherapeutic agent mitoxantrone (Mit) was loaded to LPNPs (Mit-LPNPs) and investigated in breast cancer cells (MCF-7). This nanoformulation showed an antitumor effect, suggesting that LPNPs enhance the therapeutic efficacy of the drug Mit and effectively overcome the barriers formed by tumors, the latter being investigated in a 3D in vitro cancer model [91]. Another study investigated the efficacy of LPNPs loaded with a folic acid-doxorubicin conjugate (FAD-LPNPs) on brain cancer cells. This conjugate exerted a cytotoxic effect on cancer cells, which was mainly due to the fact that FAD-LPNPs had a higher uptake capacity via folic acid receptors that were overexpressed in cancer cells. Once it entered the cell, the folic acid–doxorubicin conjugate was hydrolyzed and doxorubicin was released into the cell, thereby exerting its cytotoxic effect [92].

In recent years, the application of LPNPs has focused on RNA-based therapies, which are considered to have therapeutic potential in the treatment of cancer due to their ability to silence key genes in tumor progression [93]. In this sense, a co-delivery system for Bcl-2 targeted agents was developed in one study. This system of interfering RNA (siRNA) conjugated with doxorubicin on LPNPs was investigated in Burkitt lymphoma (Raji) cells. The LPNPs system silenced Bcl-2 while doxorubicin was released in a controlled manner in the tumor cells, resulting in an inhibitory effect on tumor growth mediated mainly by apoptosis in an in vivo model. Doxorubicin has also been shown to enhance Bcl-2-targeted RNAi therapy [94].

Furthermore, in one study, methoxy poly(ethylene glycol)-poly(caprolactone) hibridized with dimethyldioctadecylammonium bromide (DDAB) cationic lipid (mPEG-PCL-DDAB) nanoparticles were generated and used as carriers of lycopene and insulin-like growth factor 1 receptor silencing by siRNA. The antitumor effect of this nanocarrier was investigated in breast cancer cells (MCF-7). A cytotoxic effect was shown by increasing the apoptosis rate as well as an arrested cell cycle in the cancer cells. This indicates a synergistic effect between lycopene and insulin-like growth factor 1 receptor silencing [95].

Lipid polymer hybrid nanoparticles (LPHNPs) are another drug carrier system used in cancer treatment. These NPs have a core–shell structure as they can encapsulate both hydrophobic and hydrophilic drugs, including nucleic acid (siRNA), in the core, while the shell forms a barrier that increases the release time of the drug, making these systems highly therapeutically effective [96]. One study demonstrated the efficacy of LPHNPs in human prostate cancer cell lines (prostate-specific membrane antigen (PSMA) positive LNCaP cells and PSMA-negative PC3 cells) and in vivo. For this study, an aptamer-functionalized curcumin and cabazitaxel co-delivered LPHNPs (APT-CUT/CTX-LPHNPs) was developed. This nanosystem showed a better inhibitory effect on cancer cell growth in the LNCaP cell line as well as increased cellular uptake, which showed higher tumor accumulation (in vivo), indicating an antitumor effect on PSMA-expressing cells [97].

In addition, these LPHNPs can be functionalized with a cancer cell-specific ligand so that they can penetrate the cells. By combining drug and controlled release, these NPs can be used in the treatment of various cancers such as breast, lung, prostate, skin, blood, brain, liver cancer and others [98,99]. In this sense, a study developed a nanostructured therapeutic system consisting of epidermal growth factor (EGF)-functionalized LPHNPs co-loaded with 5-fluorouracil and sulforaphane (EGF-LPNPs/5-FU-SFN) and tested it on colon carcinoma cells (HCT-15). These EGF-functionalized LPNPs showed increased uptake by cancer cells, mainly due to EGF overexpressed by HCT-15 cells, and a higher inhibition rate of cell viability, leading to a greater cytotoxic effect of EGF-LPNPs/5-FU-SFN on cancer cells [100].

Another study showed the efficacy of LPNPs conjugated with a peptide (GE11) and the chemotherapeutic agent salinomycin. The peptide had a high affinity for the epidermal growth factor receptor (EGFR), which conferred a greater targeting effect on the EGFR, which was overexpressed on the surface of cancer cells. This nanosystem (GE11-LPNPs-SAL) was studied in vivo on human osteosarcoma cells (U2OS). These NPs showed a cytotoxic effect as well as enhanced uptake into cancer cells, they also suppressed osteosarcoma cell migration and decreased tumor volume in the in vivo model, suggesting an antitumor effect of GE11-LPNPs-SAL in osteosarcoma cells [101].

Combined drug therapy is necessary when dealing with an aggressive cancer such as colorectal cancer, where the first-line treatment is the combination of 5-fluorouracil, irinotecan and leucovorin (FOLFIRI), but this treatment is considered very toxic and adversely affects the patient. The application of targeted systems is necessary, such as the use of polydopamine nanoparticles PDA-NPs loaded with FOLFIRI and functionalized with an anti-EGFR antibody (cetuximab), an antibody against CRC cells overexpressing EGFR (FOLFIRI-CTX@ PDA-NPs). These were investigated in colon carcinoma cells (HTC116 and HT29). The cancer cells internalized FOLFIRI-CTX@ PDA-NPs, mainly due to the presence of anti-EGFR antibodies, suggesting that the internalization of the nanosystem occurred via receptor-mediated endocytosis process and showed a therapeutic effect in vitro, as it significantly reduced the viability of HTC116 and HT29 cells [102]. Therefore, the development of functionalized nanocarriers may increase the therapeutic efficacy of anticancer drugs and reduce the side effects of these drugs when a combination of drugs is required to treat aggressive cancers.

#### 3.2.3. Metal–Organic Frameworks

Metal–organic frameworks (MOFs) are structures produced by combining organic ligands with metal ions to produce crystalline materials with high porosity and large surface area [103]. Porosity is one of their most important properties as it can be modulated and controlled using appropriate ligands, metal centers and various environmental conditions such as pH, temperature, reaction time, pressure, etc. [104]. Due to these properties, they are of great interest especially for drug delivery as they can be easily functionalized to adjust porosity, shape, size and chemical composition [105]. A wide range of drugs can be incorporated into MOFs, whether they are hydrophilic, hydrophobic or amphiphilic. The binding of drugs can be covalent or non-covalent, depending on the type of drug and the target site [106]. In cancer treatment, MOFs as drug delivery systems can enhance drug release and increase the bioavailability of drugs at the target site. These systems also enable the exploration of multimodal treatments that can be used to treat cancer cells that are resistant to conventional treatments [107].

In this framework, a metal–organic framework-encapsulated drug delivery NPs (Zeolitic imidazolate frameworks-8) with paclitaxel modified with di-peptide (WQPDTAHHWATL) was designed to increase its specificity in human prostate cancer cells (Lncap). The results of this study showed that the MOFs entered the cancer cells by receptor-mediated endocytosis, mainly due to their surface modification with the di-peptide. Once inside the cells, the MOFs exerted their cytotoxic effect, which was mainly mediated by apoptosis in the cancer cells. It has also been observed that this nanosystem can effectively inhibit the migration and invasion ability of Lncap cells, thus suppressing tumor metastasis [108]. In another study, a novel drug carrier system consisting of iron-based MOFs (MIL-101) and graphene oxide (GO) co-loaded with luteolin and matrine was developed. The efficacy of this carrier was studied in human colorectal cancer (RKO) cells and showed a cytotoxic effect through increased ROS release and upregulation of caspase-3 and caspase-9, as well as the inhibition of RKO cell migration, suggesting that this complex drug carrier system has an anticancer effect by combining the action of both luteolin and matrine. The last molecule helped to regulate the pH of the system, stabilized the carrier and served as an excipient to enable the uptake of both drugs [109].

In a recent study, copper-based metal–organic frameworks (Cu-MOFs) with a saccharide such as heparin (Hep) and loaded doxorubicin (Cu-MOFs/Hep-DOX) were developed and investigated in human breast cancer cells (MCF-7). The results showed that the Cu-MOFs/Hep-DOX exhibited a decrease in viability accompanied by nuclear fragmentation and chromatin condensation. This suggests that the use of active targeted nanomedicine for anticancer drug delivery improves drug release as well as the efficacy of the anticancer drug [110].

In addition, GSH-sensitive prodrug nano-MOFs (nMOFs) were synthesized by the coordination of CuCl_2_, doxorubicin was encapsulated into the nanoporous structure and the surface of the nMOFs was functionalized with hyaluronic acid (nMOF-DOX-HA) to determine the anticancer effect; the nMOFs were examined in vitro (HepG2 cells, Hela cells, U87MG cells and 4T1 cells) and in vivo (H22 cells). This system enabled specific uptake in HepG2 cells overexpressing the CD44 receptor. Once inside the cells, the nMOF-DOX-HA was degraded by the acidic microenvironment and the presence of high concentrations of GSH-H_2_O_2_, releasing doxorubicin and thus exerting a cytotoxic effect on HepG2 cells, this effect being mainly mediated by apoptosis. On the other hand, in vivo studies showed a decrease in tumor growth and increased AMPK enzyme activity in tumor tissue, which promotes chemotherapeutic efficacy [111].

### 3.3. Carbon-Based Nanomaterials

Carbon is considered a biocompatible element because several of the biomolecules that make up the body are composed of this element, for example, carbohydrates, proteins, deoxyribonucleic acid (DNA) and lipids. Carbon-based nanomaterials (CBNMs) are structures made from carbon. Therefore, some of their structures are considered biocompatible and potentially used in biomedical applications such as imaging, biosensors, tissue engineering, drug delivery and cancer treatment [112], as shown in Table 3.

Carbon-based nanomaterials such as nanotubes and graphene have unique properties such as thermal conductivity, large surface area and electrical properties, which is why they are considered interesting molecules in the field of phototherapy against cancer. This therapy is a therapeutic modality that includes the diagnosis, targeting and treatment of cancer [113]. Within phototherapy, there is photodynamic therapy (PDT) and photothermal therapy (PTT). PDT is a non-invasive cancer treatment that relies on the excitation of photosensitizers (PS) at a specific wavelength to generate ROS such as singlet oxygen to kill cancer cells by oxidizing cell components, such as the cell membrane, proteins and DNA [114], to eventually lead the cell to apoptosis, necrosis or autophagy [115] (Figure 3).

PDT is more selective than standard treatments, as the PS accumulate in the tumor and the effect of the radiation is therefore limited to the tumor. This reduces side effects, and there is no drug resistance [116]. In order to achieve a greater effect with PDT and PTT, it has been proposed to combine the two therapies by coupling a highly absorbent nanosystem with a PS. The application of nanosystems based on multi-walled carbon nanotubes (MWCNT) and the PS m-tetrahydroxyphenylchlorine (MWCNT-mTHPC) in PDT and PTT was investigated in SKOV3 ovarian cancer cells. This combined therapy induced oxidative stress-mediated apoptosis and mitochondrial damage in SKOV3 cells. In response to this damage, the cell generated additional ROS via the mitochondria, which slightly increased the redox-sensitive transcription factor. On the other hand, some antioxidant enzymes and anti-apoptotic proteins were decreased, including the apoptosis inhibitors cIAP-1 and survivin, so that PDT and PTT induced apoptosis via intrinsic pathways. Overall, the cytotoxic effects of the combined treatment of PDT/PTT induced a high rate of apoptosis compared to PDT or PTT, demonstrating the efficacy of the combined therapy [117].

Carbon nanotubes (CNTs) and graphene play an important role as drug delivery systems, as they have a large surface area that enables a high loading capacity of drugs, biomolecules or PS [118]. CNTs are poorly soluble and dispersible, which limits their potential applications. However, an improvement of their properties through surface modification is necessary to increase their solubility and biocompatibility, deepen penetration and reduce their toxicity in biological systems [119]. The modification of CNTs is carried out using physical or chemical methods. Physical methods include π-π interaction, H-bonding and electrostatic interaction. Chemical methods include three grafting methods: “grafting on, grafting off and grafting through”, which are used in the functionalization of carbon-based nanomaterials [120,121]. Functionalization of CNTs can increase the efficacy of antitumor drugs and limit the toxic effects on non-cancer cells [122]. The functionalization of CNTs with small peptides can increase the specificity of these materials for cancer cells. One example is arginylglycylaspartic acid (RGD), a peptide that serves as a ligand for various integrins that are overexpressed in some cancers [123]. Recently, a study investigated the effect of a cyclic RGD peptide conjugated to the surface of carboxylic acid-functionalized carbon nanotubes (fCNTs). The topoisomerase I inhibitor camptothecin (CPT) was encapsulated in the fCNTs (CPT-fCNT-RGD), and these were examined on melanoma (A375) and breast cancer cells (MCF-7). When the A375 cells were exposed to CPT-fCNT-RGD, they showed a decrease in cell viability, while MCF7 cells showed no difference in cell viability. This effect was mainly due to the fact that A375 cells expressed integrin αvβ3, whereas MCF7 cells did not. This proves a selective cytotoxic effect on cancer cells expressing integrin αvβ3. Furthermore, the cytotoxic effect of CPT-fCNT-RGD was shown to result from the increase in expression of caspase-3, NF-kB and Bax [124]. The same cytotoxic effect was demonstrated with RGD-decorated chitosan (CS)-functionalized single-walled carbon nanotubes (SWCNT) carriers using docetaxel (RGD-SWCNT-DTX) on A549 cells, uptake was also higher, but this effect was not observed in MCF7 cells. When the anticancer effect was evaluated in vivo, this nanostructure inhibited tumor growth in a specific manner, suggesting that RGD-SWCNT-DTX could release the drug at the tumor, reducing side effects and increasing specificity due to the interaction of RGD and the integrin αvβ3 expressed by the tumor cell [125].

Another important method in cancer treatment is through the epidermal growth factor receptor, which is overexpressed in some tumor cells [126]. Therefore, functionalizing CNTs with epidermal growth factor (EGF) gives these materials the ability to target tumor cells and thus improve the therapeutic efficacy of anticancer drugs. In one study, researchers designed a targeted drug delivery system (TDDS) with the aim of increasing the efficacy of the drug etoposide (ETO). This design was developed using epidermal growth factor–chitosan–carboxyl single-walled carbon nanotubes–ETO (EGF-CHI-SWNT-COOHs-ETO) and evaluated on A549 cells. This revealed a cytotoxic effect mediated mainly by apoptosis and necrosis. This suggests that TDDS must target cancer cells overexpressing the EGF receptor and that ETO was released and entered the cells to exert its cytotoxic effect, demonstrating the efficacy of this system as a drug carrier [127]. In another study, a carrier system for the drug 7-ethyl-10-hydroxycamptothecin (SN38) was developed. Single-walled carbon nanotubes (SWCNTs) functionalized with cetuximab (C225), a chimeric (mouse/human) monoclonal antibody that is an inhibitor of the EGF receptor (EGFR), were used for targeted therapy of EGF receptor-overexpressing colorectal cancer cells (HCT116 and HT29) and SW620 cells. In this study, the effect of SWCNTs functionalized with C225 and the drug SN28 (SWCNTs-C225-SN28) was demonstrated. They had a cytotoxic effect on HCT116 and HT29 cells overexpressing EGFR, mainly through apoptosis and necrosis, while cells not expressing this receptor showed no cytotoxic effect. This suggests that the conjugation of SWCNTs with C225 directs the nanomaterials to the cancer cells and SN28 is released, which could be a good target vehicle in cancer therapy and thus exerts its cytotoxic effect [128].

Graphene (GFN), on the other hand, has the ability to interact with cell membranes and cause damage by generating ROS. It has been observed that this nanomaterial accumulates in tumors due to its shape and has a longer circulation lifetime compared to CNTs [129]. Like CNTs, GFN can be functionalized, which affects the surface charge and hydrophobicity and alters the degree of ionization, ultimately changing the biological effect. Biological applications of this material include drug/gene delivery, photothermal cancer therapy, imaging and biosensing [130]. The functionalization of GFN with different molecules such as DNA, proteins or polymers or with different functional groups can increase biocompatibility and circulation lifetime and give the material additional properties for a specific application, e.g., for targeting a specific cell or the specific treatment of diseases such as cancer [131]. Due to the various functionalization strategies of GFN, it can be used as a drug delivery system with controllable biological properties. Graphene polymer platforms have been used in the development of new drug delivery systems due to their high drug loading capacity [132]. In one study, a PEGylated nano-ghapene oxide (pGO)-based drug delivery system for cisplatin (Pt) and DOX (pGO-PT/DOX) was developed. This system was evaluated in human cell carcinoma cell lines (CAL-27) and MCF-7 cells. It was shown that this nanosystem had a cytotoxic effect in the cells through both apoptosis and necrosis. In the in vivo study, pGO-PT/DOX was observed to be released into tumor cells as larger amounts of the drug accumulated in the tumor compared to the Pt/DOX mixture alone, which led to a reduction in tumor size, suggesting that the dual-release system accumulated more efficiently in the tumor and suppressed tumor cell proliferation [133]. This more effective anticancer effect is mainly due to the different mechanisms exerted by the drugs. For example, Pt has a DNA-damaging effect that triggers senescence or apoptosis [134]. Meanwhile, the effect of DOX on cancer cells is based on the inhibition of DNA replication, which causes an inhibition of proliferation and thus exerts a synergistic effect in the treatment of cancer [135]. Another study designed a prodrug consisting of the functionalization of graphene oxide (GO) bound to the anticancer drug DOX and the copolymers PEG-polycaprolactone linked by a redox-sensitive disulfide (GO-PEG-PCL-SS-DOX). This system was evaluated in A549 cells and murine malignant melanoma cells (B16), with in vitro cytotoxicity showing that the release of the drug DOX affecting A549 and B16 cells had a more efficient anticancer effect. In addition, in the in vivo study, this nanosystem showed an antitumor effect by significantly reducing the tumor volume, suggesting that there is a high accumulation in the cancer cells and that the release of DOX is controlled, reducing the toxic effects on other tissues, such as the heart [136].

Upon irradiation, GNF can absorb photons and transfer the absorbed energy to the molecules around it, which can be oxygen or some PS. This energy transfer process promotes the formation of ROS. Based on this effect, these materials can be used as PDT and PTT for cancer [137]. To combat drug resistance, the surface of GNF was modified with PEG and oxidized sodium alginate (OSA), which is pH-sensitive, and then loaded with paclitaxel (PTX) to generate a drug delivery system (PTX-GO-PEG-OSA). This system was tested in PTX-resistant GC cells (HGC-27/PTX) and demonstrated the therapeutic efficacy of PTX by reducing cell proliferation and increasing apoptosis. This increase in apoptosis was associated with the expression of caspase-3, BAX and p53 proteins, while anti-apoptotic markers such as BCL-2 were suppressed. In addition, PTX-GO-PEG-OSA with NIR was shown to exhibit increased cytotoxicity compared to the cells not irradiated with NIR, indicating a photothermal effect that generates ROS and increases PTX release. This demonstrates the therapeutic efficacy of this drug delivery system in drug-resistant cancers [138]. GNF induces ROS production in the cell. This is a mechanism by which it can exert a cytotoxic effect on cancer cells. To evaluate this effect, GNF molecules were functionalized with trastuzumab (anti-HER2 antibody), yielding GFN-TRA. This nanomaterial was evaluated in osteosarcoma cells (MG63, HOS and 143B) expressing low levels of HER2 and in an in vivo model. GNF-TRA induced ROS production as well as HER2 signaling, which ultimately led to cell death mediated mainly by necroptosis. In the in vivo model, both tumor eradication and lung metastasis were achieved, indicating an effective treatment in cancer treatment [139]. In another study, a PTX-GO-VEGF system was developed consisting of a PTX-loaded GO encapsulated in human serum albumin (HSA) and functionalized with monoclonal antibodies (mAb) against vascular endothelial growth factor to form a nanodrug. This nanodrug was functionalized via a PEG linker with an antibody against vascular endothelial growth factor (VEGF). This system was tested in human adrenocortical carcinoma cells (SW-13) and in a solid tumor in mice. Targeted photothermally induced tumor chemotherapy showed a high cell death rate, but even when the cells were irradiated with NIR, cell death increased. This effect is mainly due to the fact that NIR irradiation triggers hyperthermia, which leads to an increased release of PTX in the tumor cells and thus improves chemotherapy [140]. The development of multifunctional systems based on GO as well as drug delivery systems with on/off properties is a system that represents an alternative for the treatment of cancer. Based on this, a nanodrug was developed, which was GO functionalized with a monoclonal integrin αvβ3 antibody and conjugated with pyropheo-phorbide-a (PPa) and PEG (PPA-PEG-GFN-mAb). This system was investigated in adherent human glioblastoma cells (U87-MG) and MCF-7 cells. This system showed a cytotoxic effect specifically in U87-MG cells expressing integrin αvβ3. This integrin mediated entry into the cell and was directed to the mitochondria, where the “switched-on” PPA-PEG-GFN-mAb exerted its phototoxic effect to increase mitochondrial-mediated apoptosis in cancer cells [141].

**Table 1 ijms-26-02633-t001:** Applications of functionalized gold, platinum and silver nanoparticles in cancer therapy.

Nanoparticle Type	Functionalization Material	Type of Cancer	Mechanism of Action	Combined Radiotherapy	Reference
AuNPs	2-thiouracil	Breast cancer cells (MDA-MB-231)	Antiproliferative activity of 2-TU and PTT effect	No	[45]
AuNPs	HER-2 and dasatinib	Breast cancer cells (BT-474 and MCF-7)	IIncrease in the activity of dasantinib	Yes	[46]
AuNPs	Trastuzumab	Gastric cancer cells (MKN7 and NCI-N87) and in vitro and vivo; melanoma cells (G361)	Autophagy	No	[47]
AuNPs	Anti-HER2	melanoma cells (G361)	Apoptosis and cell cycle arrest	No	[48]
PtNPs	PEG-coated and doxorubicin	lung cancer cells (A549) in vitro and in vivo	Apoptosis and cell cycle arrest	No	[53]
PtNPs	Doxorubicin-conjugated PtNPs in octopod form	Breast cancer cells (MCF-7 and MDA-MB-231)	Mitochondrial dysfunction and activation of caspases-3 and -9 (Apoptosis)	No	[54]
PtNPs	PEG-coated and doxorubicin	Breast cancer cells (MCF-7/ADR)	Combination of chemotherapy and phototherapy	No	[55]
PtNPs	Chitosan	Breast cancer cells (MDA-MB-231; MCF7)	Apoptosis	No	[57,58]
PtNPs	Lactic-co-glycolic acid, PEG, anti-EGFR	Breast cancer cells (MDA-MB-231)	Oxidative state of PtNPs	No	[59]
PtNPs	Hyaluronic acid	Breast cancer cells (MDA-MB-231) in vitro and in vivo	Photothermal therapy	No	[60]
AgNPs	Glucose	Prostate cancer cells (DU-145, PC-3 and LNCaP)	Oxidative damage, DNA fragmentation, apoptosis and cell cycle arrest	No	[65]
AgNPs	Polyvinyl alcohol, PGE, polyvinylpyrrolidone and conjugated with doxorubicin	Breast cancer cells (MCF-7)	Oxidative stress and lipid peroxidation	No	[67]
AgNPs	Chitosan	Breast cancer cells (MCF-7)	DNA damage, mitochondrial damage and apoptosis	No	[68,69]
AgNPs	PGE, folic acid and conjugated with doxorubicin	Adenocarcinoma; lymphocytic leukemia	Generation of reactive oxygen species	No	[71]
AgNPs	Anti-EGFR	Nasopharyngeal carcinoma	Apoptosis	Yes	[72]
AgNPs	IgG	Pancreatic cancer cell (Panc-1)	Apoptosis	No	[73]

**Table 2 ijms-26-02633-t002:** Functionalized PLGA-based nanoparticles, lipid nanoparticles and metal–organic frameworks in cancer therapy.

Nanoparticle Type	Functionalization Material	Type of Cancer	Mechanism of Action	Reference
PEG-PLGA	anti-CD133; methioninase; pemetrexed	Gastric cancer cells (CD133+ SGC7901 and MKN45)	Apoptosis a and inhibition of DNA synthesis	[81]
PLGA	anti-CD133; oxaliplatin	Colorectal cancer (Caco-2)	Inhibition of the DNA synthesis	[82]
Lipid-PLGA	Anti-CD44; salinomycin	CD44+ prostate cancer cells (DU145 and 22RV1)	-	[83]
PLGA	Anti-EGFR; paclitaxel	Triple-negative breast cancer (MDA-MB-468 TNBC) and in vivo	Condensed chromatin, fragmented nucleus and formation of apoptotic bodies	[85]
PLGA	Antibodies against Frizzled7; doxorubicin	Triple-negative breast cancer (MDA-MB-231)	Apoptosis and/or necrosis	[86]
LPNPs	methotrexate	Human lung cancer cell (A549)	Apoptosis	[89]
LPNPs	mitoxantrone	Breast cancer (MCF-7)	-	[90]
LPNPs	Folic acid; doxorubicin	Brain cancer (U87 MG)	-	[91]
LPNPs	Doxorubicin; siRNA	Burkitt lymphoma (Raji)	Apoptosis	[93]
mPEG-PCL-DDAB	Lycopen; insulin-like growth factor 1 receptor siRNA	Breast cancer (MCF-7)	Apoptosis and arrested cell cycle	[94]
LPHNPs	Curcumin; cabazitaxel	Prostate cancer (LNCaP and PC3) and in vivo	-	[96]
LPHNPs	EGF; 5-fluorouracil; sulforaphane	Colon carcinoma (HCT-15)	Apoptosis	[99]
LPHNPs	Polypeptide GE11; salinomycin	Osteosarcoma (U2OS) and in vivo	Suppress the migration and proliferation	[100]
Polidopamine nanoparticles	Cetuximab; 5-fluorouracil; irinotecan; leucovorin	Colon carcinoma (HTC116 and HT29)	-	[101]
MOFs	Di-peptide (WQPDTAHHWA-TL); paclitaxel	Prostate cancer (Lncap)	Apoptosis	[107]
Iron-based MOFs	Graphene oxide; luteolin and matrine	Colon cancer (RKO)	ROS, upregulated caspase-3 and caspase-9 and inhibition in the migration	[108]
Copper-based MOFs	Heparin; doxorubicin	Cancer breast (MCF-7)	Nuclei fragmentation and chromatin condensation	[109]
nMOFs	Hyaluronic acid; doxorubicin	HepG2 cells, Hela cells, U87MG cells, and 4T1 cells and in vivo	Apoptosis	[110]

**Table 3 ijms-26-02633-t003:** Functionalized carbon-based nanomaterials for cancer therapy.

Nanoparticle Type	Functionalization Material	Type of Cancer	Mechanism of Action	Combined Radiotherapy	Reference
Multi-walled carbon nanotubes	m-tetrahydroxyphenylch-lorine	Ovarian cancer (SKOV3)	Oxidative stress; apoptosis and mitochondrial damage	PDT; PTT	[117]
Carbon nanotubes	Arginylglycylaspartic acid; camptothecin	Melanoma and breast cancer (A375 and MCF7)	Increase in expression of caspase-3, NF-kB and Bax	No	[124]
Single-walled carbon nanotubes	Arginylglycylaspartic acid; camptothecin; chitosan; docetaxel	Lung cancer, breast cancer (A549 cells and MCF-7) and in vivo	-	No	[125]
Carbon nanotubes	Epidermal growth factor; chitosan and etoposide	Lung cancer (A549)	Apoptosis and necrosis	No	[127]
Single-walled carbon nanotubes	Cetumixab; 7-ethyl-10-hydroxycamptothecin	Colorectal cancer (HCT116, HT29 and SW620)	Apoptosis and necrosis	No	[128]
Ghapene oxide	PEG; cisplatin; doxorubicin	Squamous cell carcinoma and breast cancer (CAL-27) and MCF-7)	Apoptosis and necrosis	No	[133]
Ghapene oxide	PEG-polycaprolactone; doxorubicin	Lung cancer and skin cancer (A549 and B16)	Necrosis	No	[136]
Graphene	PEG; oxidized sodium alginate; paclitaxel	Paclitaxel-resistant gastric carcinoma cell (HGC-27/PTX)	Apoptosis, oxidative stress	PTT	[138]
Graphene	Trastuzumab	Osteosarcoma (MG63, HOS, 143B)	Oxidative stress, necroptosis	No	[139]
Graphene oxide	Antibodies against vascular endothelial growth factor; HAS; paclitaxel	Adrenocortical carcinoma (SW-13) and in vivo	Apoptosis	PTT	[140]
Graphene oxide	Integrin αvβ3 antibody; pyropheo-phorbide-a; PEG	Glioblastoma and breast cancer (U87-MG, MCF-7 cells.	Apoptosis	PDT	[141]

## 4. Future Perspectives

Nanotechnology offers the possibility of using innovative treatments in cancer therapy. There are currently clinical trials of nanoparticles in cancer treatment, such as carbon, iron and hafnium oxide nanoparticles, liposomes, as well as target-enhancing nanoparticles of paclitaxel and other nanomaterials. Some of these studies have already reached phase 3 and even phase 4, suggesting that these nanosystems are promising in cancer treatment [142]. In recent years, in vitro and in vivo results have shown the efficacy of treatments with these nanomaterials. Among the phase 3 studies, the application of carbon nanoparticles in patients with thyroid cancer stands out [143]. Functionalized hafnium oxide nanoparticles in combination with cetuximab were implemented for the treatment of squamous cell carcinoma of the head and neck [144]. However, a growing number of trials are being developed, with some in phase 1 and others in phase 2 for some nanomaterials that are expected to eventually be used in patients and thus provide a new treatment against cancer.

## 5. Conclusions

NPs are promising nanomaterials for cancer treatment as they can enable targeted therapies and the controlled release of drugs. Nanomaterials have unique properties, mainly due to their high surface/mass ratio, small size, composition, unique surface reactivity and biocompatibility. These properties can improve cancer therapies as NPs can be used in combination therapy (thermal, radiation and light), multidrug therapy (drugs, protein drugs) and targeted drug delivery (antibodies, peptides and biomolecules), thus increasing their therapeutic efficacy on cancer cells. The cytotoxic effects that NPs have on cancer cells range from the generation of ROS, apoptosis, necrosis and cell cycle arrest to the suppression of anti-apoptotic genes. These effects depend on the type of material with which they were synthesized, the biomolecule with which they were functionalized and the drug carried by the NPs. This is why these nanomaterials are currently an innovative strategy in the development of cancer therapies.

## Figures and Tables

**Figure 1 ijms-26-02633-f001:**
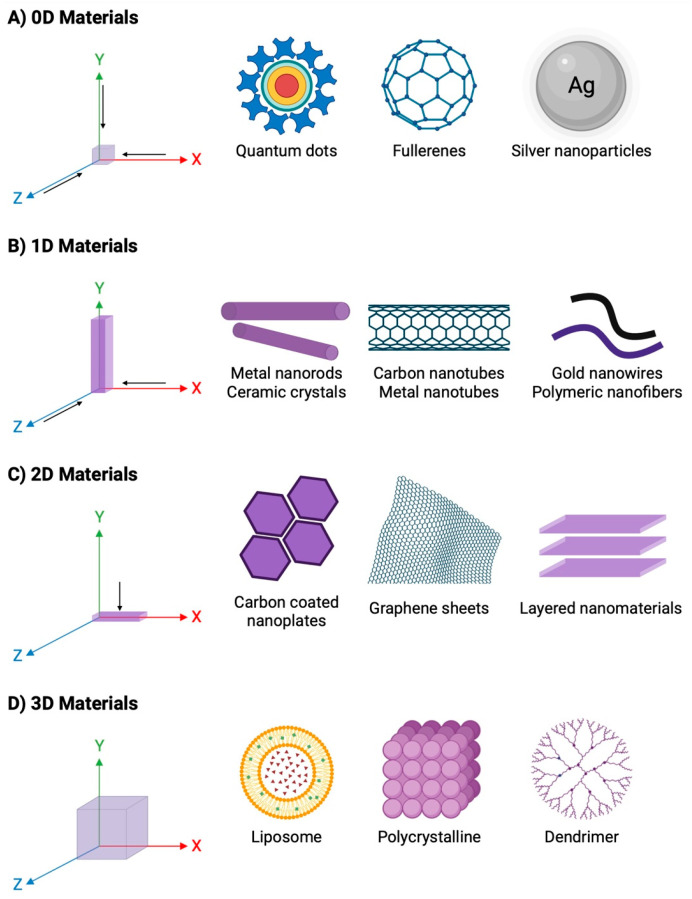
Classification of nanomaterials based their dimensionality. (**A**) 0D Materials: quantum dots, fullerenes and silver nanoparticles; (**B**) 1D Materials: metal nanorods, ceramic crystals, carbon nanotubes, metal nanotubes, and gold nanowires; (**C**) 2D Materials: carbon coated nanoplates, graphene sheets and layered nanomaterials; and (**D**) 3D Materials: liposome, polycrystalline and dendrimer. Figure created with https://www.biorender.com (accessed on 4 November 2024).

**Figure 2 ijms-26-02633-f002:**
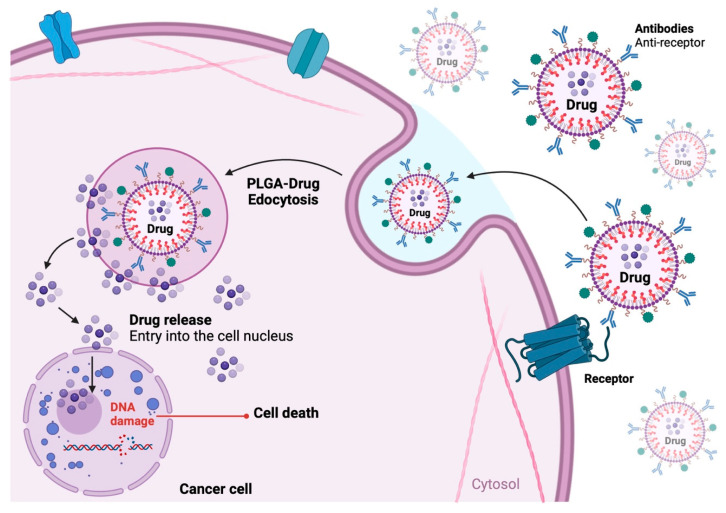
Effect of anticancer drugs encapsulated in PLGA and functionalized with an antibody. Figure created with https://www.biorender.com (accessed on 4 November 2024).

**Figure 3 ijms-26-02633-f003:**
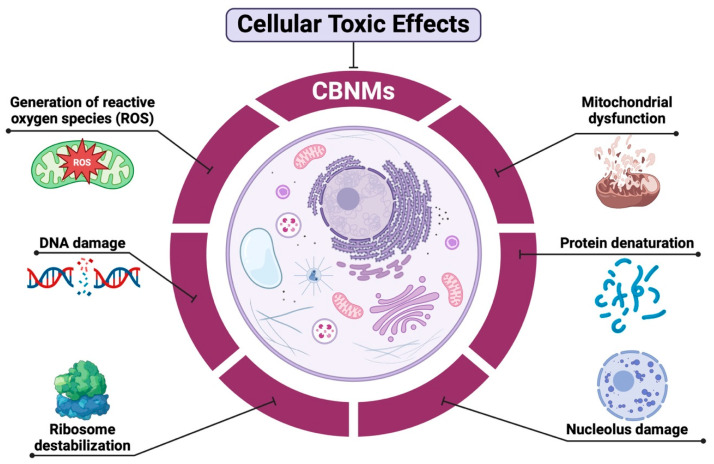
Effect of carbon-based nanomaterials on mitochondria, DNA, ribosomes, proteins and the nucleus of cancer cells. Figure created with https://www.biorender.com (accessed on 4 November 2024).

## Data Availability

Not applicable.

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
