# Peer review of "Functionalized Nanomaterials in Cancer Treatment: A Review"

_ijms, 2025, doi:10.3390/ijms26062633_

Round 1
Reviewer 1 Report
Comments and Suggestions for Authors
Oscar Gutiérrez Coronado et al. prepared a review on cancer treatment using functional nanomaterials. Overall, the manuscript is in good shape; however, a few concerns need to be addressed before acceptance:
- The title is somewhat broad for this review paper.
- It is recommended to include more figures and tables to enhance clarity.
- Consider restructuring the manuscript to differentiate between the materials section and the applications section clearly.
- If possible, include a wider variety of nanomaterials to improve the manuscript's logical flow.
- Lastly, please double-check the formatting of the references.
Author Response
Based on your comments and the observations of the other referee, we have modified the manuscript.
Point 1: The title is somewhat broad for this review paper.
Response 1: Thank you for your comment. We think the title is appropriate, but if the editor allows us to change it, we would opt for the following title: “Metallic, organic and carbon-based nanomaterials: functionalization and applications in cancer therapy”.
Point 2: It is recommended to include more figures and tables to enhance clarity.
Response 2: The manuscript was modified based on the basis of the commentaries and Tables 1, 2 and 3 have been inserted.
Point 3: Consider restructuring the manuscript to differentiate between the materials section and the applications section clearly
Response 3: Sections 3.1. Metallic nanoparticles, 3.2. Organic nanoparticles and 3.3 Carbon-based nanomaterials have basically the same format, in which a brief introduction to the nanomaterial is given, followed by its applications in therapy against various types of cancer. We consider this appropriate as several articles follow this format.
Point 4: If possible, include a wider variety of nanomaterials to improve the manuscript's logical flow.
Response 4: Section 3.2 Organic nanoparticles was expanded in manuscript lines 436-566.
Point 5: Lastly, please double-check the formatting of the references.
Response 5: The references were checked and corrected.
Dear referee, we greatly appreciate all your comments, we hope that our answers are adequate, so that you consider our article for publication. Best regards.

Reviewer 2 Report
Comments and Suggestions for Authors
While the manuscript provides some useful information, it primarily summarizes existing studies without offering significant new insights into the field. Given the abundance of similar reviews on this topic, the authors should enhance the content to include more in-depth discussions. Below are my detailed comments:
-
The organization needs improvement. Section 3.2, titled "Organic Nanoparticles," should be subdivided into more specific sections such as polymeric nanoparticles, lipid nanoparticles, and metal-organic frameworks (MOFs). Most clinically advanced formulations fall under this category, and the authors should conduct a more comprehensive literature review for each type.
-
The manuscript overlooks recent breakthroughs in the field. For example, lipid nanoparticles (LNPs) have made significant advancements in biomacromolecule delivery and have opened new therapeutic avenues, including gene therapy and gene editing. These developments should be thoroughly discussed.
-
The potential for clinical translation of these formulations should be addressed. What are the major challenges that need to be overcome before clinical application? Which formulations have already demonstrated success in clinical trials, and what are the results from recent trials of representative formulations?
-
The authors should provide a more critical analysis of the formulations discussed in the manuscript. A deeper exploration of the unique features, strengths, and limitations of each formulation will provide a more comprehensive understanding of their potential.
There are many typos and grammar mistakes in the manuscript.
Author Response
Based on your comments and the observations of the other referee, we have modified the manuscript.
Point 1: The organization needs improvement. Section 3.2, titled "Organic Nanoparticles," should be subdivided into more specific sections such as polymeric nanoparticles, lipid nanoparticles, and metal-organic frameworks (MOFs). Most clinically advanced formulations fall under this category, and the authors should conduct a more comprehensive literature review for each type.
Response 1: Section 3.2 has been expanded to include lipid nanoparticles and metal-organic frameworks (MOFs), lines 436-566.
Point 2: The manuscript overlooks recent breakthroughs in the field. For example, lipid nanoparticles (LNPs) have made significant advancements in biomacromolecule delivery and have opened new therapeutic avenues, including gene therapy and gene editing. These developments should be thoroughly discussed.
Response 2: Section 3.2 has been extended to include lipid nanoparticles and metal-organic frameworks (MOFs).
Point 3: The potential for clinical translation of these formulations should be addressed. What are the major challenges that need to be overcome before clinical application? Which formulations have already demonstrated success in clinical trials, and what are the results from recent trials of representative formulations?
Response 3: A review was conducted at https://clinicaltrials.gov/, with the search limited to cancer and nanoparticles. Studies in phases 1, 2, 3 and 4 were found and we have cited 2 of the most relevant studies that are still recruiting patients (references 143 and 144). These have been included in section 4. Future prospects
Point 4: The authors should provide a more critical analysis of the formulations discussed in the manuscript. A deeper exploration of the unique features, strengths, and limitations of each formulation will provide a more comprehensive understanding of their potential.
Response 4: Section 4 was included on the basis of the commentary. The future perspectives have been inserted: Lines 735-746. Studies on carrier systems, immunotherapy, chemotherapy and photothermal therapy have been included in the manuscript and are summarized in tables.
Dear referee, we greatly appreciate all your comments, we hope that our answers are adequate, so that you consider our article for publication. Best regards.

Reviewer 3 Report
Comments and Suggestions for Authors
The review by Gutiérrez Coronado et al. aims to summarize recent studies in which various types of nanoparticles have been functionalized to increase therapeutic efficacy in cancer treatment.
One of the major issues is that there are already many reviews of this type (just as examples: Signal Transduction and Targeted Therapy (2023) 8:41; Cheng et al. J Hematol Oncol (2021) 14:85; Meraj et al, Personalized and Precision Nanomedicine for Cancer Treatment, 2024, p. 109-127), and the review by Coronado et al. in its current version does not add much to what is already found in the literature.
- The authors need to revise the text (and also the title), limiting the literature review to a restricted and specified number of years.
- The chapter on organic nanoparticles needs to be improved; indeed, among polymeric nanoparticles, there are not only PLGAs (which the authors focus on) but also (just as an example) polylipoic acid nanoparticles.
- A specific chapter should be included to discuss innovations in nanoparticle functionalization.
- The addition of tables summarizing key data from the main text is recommended throughout the manuscript.
Author Response
Based on your comments and the observations of the other referee, we have modified the manuscript.
Point 1: The authors need to revise the text (and also the title), limiting the literature review to a restricted and specified number of years.
Response 1: A review of the title was performed, if the editor allows, it would be changed to the following: “Metallic, organic and carbon-based nanomaterials: functionalization and applications in cancer therapy”.
A bibliographic review was performed for this manuscript using date filters (2021 to 2025), excluding the review articles, with keywords corresponding to the different sections of the manuscript.
Point 2: The chapter on organic nanoparticles needs to be improved; indeed, among polymeric nanoparticles, there are not only PLGAs (which the authors focus on) but also (just as an example) polylipoic acid nanoparticles.
Response 2: Section 3.2 has been expanded to include lipid nanoparticles and metal-organic frameworks (MOFs), lines 436-566.
Point 3: A specific chapter should be included to discuss innovations in nanoparticle functionalization.
Response 3: The metallic, organic and carbon-based nanomaterials mentioned in this manuscript are considered innovative because they are functionalized and can therefore have a stronger anticancer effect. This is mainly because nanomaterials are decorated with various structures, including antibodies, peptides, polymers, anticancer drugs, RNA and other molecules.
Point 4: The addition of tables summarizing key data from the main text is recommended throughout the manuscript.
Response 4: The manuscript was modified based on the basis of the commentaries and Tables 1, 2 and 3 have been inserted.
Dear referee, we greatly appreciate all your comments, we hope that our answers are adequate, so that you consider our article for publication. Best regards.

Round 2
Reviewer 1 Report
Comments and Suggestions for Authors
This review paper can be accepted in the current form.
Reviewer 2 Report
Comments and Suggestions for Authors
The authors have addressed all my previous comments.
Reviewer 3 Report
Comments and Suggestions for Authors
The authors have addressed the majority of the Reviewer's concerns. Their review can be published in its present form, with the newly proposed title.